# UNSUPERVISED META-LEARNING FOR REINFORCEMENT LEARNING

## ABSTRACT

Meta-learning is a powerful tool that learns how to quickly adapt a model to new tasks. In the context of reinforcement learning, meta-learning algorithms can acquire reinforcement learning procedures to solve new problems more efficiently by meta-learning prior tasks. The performance of meta-learning algorithms critically depends on the tasks available for meta-training: in the same way that supervised learning algorithms generalize best to test points drawn from the same distribution as the training points, meta-learning methods generalize best to tasks from the same distribution as the meta-training tasks. In effect, meta-reinforcement learning offloads the design burden from algorithm design to task design. If we can automate the process of task design as well, we can devise a meta-learning algorithm that is truly automated. In this work, we take a step in this direction, proposing a family of unsupervised meta-learning algorithms for reinforcement learning. We describe a general recipe for unsupervised meta-reinforcement learning, and describe an effective instantiation of this approach based on a recently proposed unsupervised exploration technique and model-agnostic meta-learning. We also discuss practical and conceptual considerations for developing unsupervised meta-learning methods. Our experimental results indicate that unsupervised meta-reinforcement learning effectively acquires accelerated reinforcement learning procedures without the need for manual task design, significantly exceeds the performance of learning from scratch, and even matches performance of meta-learning methods that use hand-specified task distributions in many environments.

## 1 INTRODUCTION

Reusing past experience for faster learning of new tasks is a key challenge for machine learning. Meta-learning methods propose to achieve this by using past experience to explicitly optimize for rapid adaptation (Mishra et al., 2017; Snell et al., 2017; Schmidhuber, 1987; Finn et al., 2017a; Duan et al., 2016b; Gupta et al., 2018; Wang et al., 2016; Al-Shedivat et al., 2017). In the context of reinforcement learning, meta-reinforcement learning algorithms can learn to solve new reinforcement learning tasks more quickly through experience on past tasks (Duan et al., 2016b; Gupta et al., 2018). Typical meta-reinforcement learning algorithms assume the ability to sample from a pre-specified task distribution, and these algorithms learn to solve new tasks *drawn from this distribution* very quickly. However, specifying a task distribution is tedious and requires a significant amount of supervision (Finn et al., 2017b; Duan et al., 2016b) that may be difficult to provide for large real-world problem settings. The performance of meta-learning algorithms critically depends on the meta-training task distribution, and meta-learning algorithms generalize best to new tasks which are drawn from the same distribution as the meta-training tasks (Finn & Levine, 2018). In effect, meta-reinforcement learning offloads some of the design burden from algorithm design to designing a sufficiently broad and relevant distribution of meta-training tasks. While this greatly helps in acquiring representations for fast adaptation to the specified task distribution, a natural question is whether we can do away with the need for manually designing a large family of tasks, and develop meta-reinforcement learning algorithms that learn only from unsupervised environment interaction. In this paper, we take an initial step toward the formalization and design of such methods.

Our goal is to automate the meta-training process by removing the need for hand-designed meta-training tasks. To that end, we introduce unsupervised meta-reinforcement learning: meta-learning from a task distribution that is acquired automatically, rather than requiring manual design of the

meta-training tasks. Developing effective unsupervised meta-reinforcement learning algorithms is challenging, since it requires solving two difficult problems together: meta-reinforcement learning with broad task distributions, and unsupervised exploration for proposing a wide variety of tasks for meta-learning. Since the assumptions of our method differ fundamentally from prior meta-reinforcement learning methods (we do not assume access to hand-specified meta-training tasks), the best points of comparison for our approach are learning the meta-test tasks entirely from scratch with conventional reinforcement learning algorithms. Our method can also be thought of as a data-driven initialization procedure for deep neural network policies, in a similar vein to data-driven initialization procedures explored in supervised learning (Krähenbühl et al., 2015). However, as indicated by Finn & Levine (2017), this procedure goes beyond simply being an initialization, and essentially learns an entire learning algorithm that is as expressive as any recurrent meta-learner.

The primary contributions of our work are to propose a framework for unsupervised meta-reinforcement learning, sketch out a family of unsupervised meta-reinforcement learning algorithms, and describe an instantiation of an algorithm from this family that builds on a recently proposed procedure for unsupervised exploration (Eysenbach et al., 2018) and model-agnostic meta-learning (MAML) (Finn et al., 2017a). We discuss the design considerations and conceptual issues surrounding unsupervised meta-reinforcement learning, and provide an empirical evaluation that studies the performance of two variants of our approach on simulated control tasks. Our experimental evaluation shows that, for a variety of tasks, unsupervised meta-reinforcement learning can effectively acquire reinforcement learning procedures that perform significantly better than standard reinforcement learning and other alternatives in terms of sample complexity and asymptotic performance, and even rival the performance of conventional meta-learning algorithms that are provided with hand-designed task distributions.

## 2    RELATED WORK

Our work lies at the intersection of meta reinforcement learning, goal generation, and unsupervised exploration. Meta-learning algorithms use data from multiple tasks to learn how to learn, acquiring rapid adaptation procedures from experience (Schmidhuber, 1987; Naik & Mammone, 1992; Thrun & Pratt, 1998; Bengio et al., 1992; Hochreiter et al., 2001; Santoro et al., 2016; Andrychowicz et al., 2016; Li & Malik, 2017; Ravi & Larochelle, 2017; Finn et al., 2017a; Munkhdalai & Yu, 2017; Snell et al., 2017). These approaches have been extended into the setting of reinforcement learning (Duan et al., 2016b; Wang et al., 2016; Finn et al., 2017a; Sung et al., 2017; Mishra et al., 2017; Gupta et al., 2018; Houthooft et al., 2018; Stadie et al., 2018), though their performance in practice depends on the user-specified meta-training task distribution. We aim to lift this limitation, and provide a general recipe for avoiding manual task engineering for meta-RL. To that end, we make use of unsupervised task proposals. These proposals can be obtained in a variety of ways, including adversarial goal generation (Sukhbaatar et al., 2017; Held et al., 2017), information-theoretic methods (Gregor et al., 2016; Eysenbach et al., 2018), and even random functions.

Methods that address goal generation and curriculum learning have complementary aims. Graves et al. (2017) study this problem for supervised learning, while Forestier et al. (2017) apply a similar approach to robot learning. Prior work (Schaul et al., 2015; Pong et al., 2018; Andrychowicz et al., 2017) also studied learning of goal-conditioned policies, which are closely related to meta-reinforcement learning in their ability to generalize to new goals at test time. However, like meta-learning, goal-conditioned policies typically require manually defined goals at training time. Although exploration methods coupled with goal relabeling (Pong et al., 2018; Andrychowicz et al., 2017) could provide for automated goal discovery, such methods would still be restricted to a specific goal parameterization. In contrast, unsupervised meta-reinforcement learning can solve arbitrary tasks at meta-test time without being restricted to a particular task parameterization. Exploration algorithms based on intrinsic motivation aim to visit diverse states (Pathak et al., 2017; Schmidhuber, 2009; Bellemare et al., 2016; Osband et al., 2016), but do not by themselves aim to generate new tasks or learn to adapt more quickly to new tasks, only to achieve wide coverage of the state space. These methods are complementary to our approach, but address a distinct problem.

Prior work has used meta-learning to learn unsupervised learning rules (Metz et al., 2018). This work learns strategies for unsupervised learning using supervised data, while our approach requires no

supervision during meta-training, in effect doing the converse: using a form of unsupervised learning to acquire learning rules that can learn from rewards at meta-test time.

Additionally, learning fast reinforcement learning algorithms unsupervised does help capture a notion of dynamics since the agent is able to interact with the world to set up tasks for itself. While a comparison here might be made with model-based RL algorithms, such a comparison is outside the scope of this work. Model-based RL algorithms generally have very different tradeoffs and assumptions than model-free methods, and while both classes are worth studying, developing more efficient model-free RL algorithms is a worthwhile endeavor, even if model-based algorithms might be more efficient: in many cases, model-free RL results in better final performance Nagabandi et al. (2017); Pong et al. (2018), and model-free algorithms typically make fewer assumptions about the underlying system.

## 3 UNSUPERVISED META-REINFORCEMENT LEARNING

The goal of unsupervised meta-reinforcement learning is to take an environment and produce a learning algorithm specifically tailored to this environment that can quickly learn to maximize reward on *any* task reward in this environment. This learning algorithm should be meta-learned without requiring *any* human supervision. We can formally define unsupervised meta-reinforcement learning in the context of a controlled Markov process (CMP) – a Markov decision process without a reward function, $C = (S, A, P, \gamma, \rho)$, with state space $S$, action space $A$, transition dynamics $P$, discount factor $\gamma$ and initial state distribution $\rho$. Our goal is to learn a learning algorithm $f$ on this CMP, which can subsequently learn new tasks efficiently in this CMP for a new reward function $R_i$. The CMP along with this reward function $R_i$ produces a Markov decision processes $M_i = (S, A, P, \gamma, \rho, R_i)$. The goal of the learning algorithm $f$ is to learn an optimal policy $\pi_i * (a|s)$ for *any* reward function $R_i$ that is provided with the CMP. Crucially, $f$ must be learned without access to any reward functions $R_i$, using only unsupervised interaction with the CMP. The reward is only provided at meta-test time. The implicit assumption in this formulation is that different tasks at test-time will all be using the same dynamics but with different reward functions. This scenario is encountered in many realistic multi-task scenarios such as a robot in a home or a factory where it has to learn many tasks in the same environment.

### 3.1 A GENERAL RECIPE

Our framework unsupervised meta-reinforcement learning consists of two components. The first component is a task identification procedure, which interacts with a controlled Markov process, without access to any reward function, in order to construct a distribution over tasks. Formally, we will define the task distribution as a mapping from a latent variable $z \sim p(z)$ to a reward function $r_z(s, a) : S \times A \rightarrow \mathbb{R}$. That is, for each value of the random variable $z$, we have a different reward function $r_z(s, a)$. The prior $p(z)$ may be specified by hand. For example, we might choose a uniform categorical distribution or a spherical unit Gaussian. A discrete latent variable $z$ corresponds to a discrete set of tasks, while a continuous representation could allow for an infinite task space. Under this formulation, learning a task distribution amounts to optimizing a parametric form for the reward function $r_z(s, a)$ that maps each $z \sim p(z)$ to a different reward function.

The second component of unsupervised meta-learning is meta-learning, which takes the family of reward functions induced by $p(z)$ and $r_z(s, a)$, and meta-learns a reinforcement learning algorithm $f$ that can quickly adapt to any task from the task distribution defined by $p(z)$ and $r_z(s, a)$. The meta-learned algorithm $f$ can then learn new tasks quickly at meta-test time, when a user-specified reward function is actually provided. This generic design for an unsupervised meta-reinforcement learning algorithm is summarized in Figure 1.

The nature of the task distribution defined by $p(z)$ and $r_z(s, a)$ will affect the effectiveness of $f$ on new tasks: tasks that are close to this distribution will be easiest to learn, while tasks that are far from this distribution will be difficult to learn. We would ideally like this distribution to be broad enough that it speeds up learning of *any* new task in expectation. However, the nature of the meta-learning algorithm itself will also crucially affect the effectiveness of $f$. As we will discuss in the following sections, some meta-reinforcement learning algorithms can generalize effectively to new tasks, while some don't do as well. A more general version of this algorithm might also use $f$ to inform the

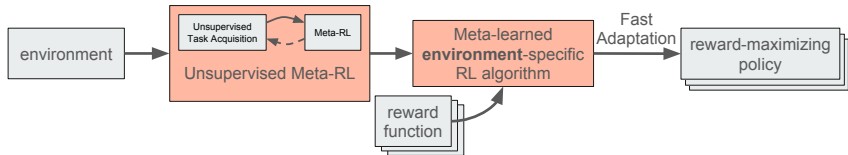

Figure 1: **Unsupervised meta-reinforcement learning**: Given an environment, unsupervised meta-reinforcement learning produces an environment-specific learning algorithm that quickly acquire new policies that maximizes any task reward function.

acquisition of tasks, allowing for an alternating optimization procedure the iterates between learning $r_z(s, a)$ and updating $f$, for example by designing tasks that are difficult for the current algorithm $f$ to handle. However, in this paper we will consider the stagewise approach, which acquires a task distribution once and meta-trains on it, leaving the iterative variant for future work.

Why might we expect unsupervised meta-reinforcement learning to acquire an algorithm $f$ that improves on any standard, generic, hand-designed reinforcement learning procedure? On the one hand, the "no free lunch theorem" (Wolpert et al., 1995; Whitley & Watson, 2005) might lead us to expect that a truly generic approach to learning a task distribution (for example, by sampling completely random reward functions) would not yield a learning procedure $f$ that is effective on any real tasks – or even on the meta-training tasks. However, the specific choice for the unsupervised learning procedure and meta-learning algorithm can easily impose an inductive bias on the entire process that *does* produce a useful algorithm $f$. For instance, in our experiments we find that DIAYN finds skills which actually move the agent meaningfully to different points in space, which is related to a large proportion of the tasks that we might care about in the environments we consider. Perhaps more importantly, even if the task distribution encompassed every possible task that could be defined, there is still benefit to the unsupervised meta-learning process in the fact that the environment dynamics are shared across tasks and would be implicitly learned and the policy would modify its behavior in ways that cogently affect the states that are visited, removing the inherent over-parameterization typically present in policy space. This means that the policy is much more likely to transition to parameters which are actually *functionally* different, when updates are performed via the meta-learned algorithm. As we will discuss below, we can identify specific choices for the task acquisition and meta-learning procedures that are generic, in the sense that they can be applied to a wide range of CMPs, but also contain enough inductive bias to meta-learn useful reinforcement learning procedures. We discuss specific choices for each of these procedures below, followed by a more general discussion of potential future choices for these procedures and the criteria that they should satisfy. We empirically validate these claims in Section 4.

### 3.2    UNSUPERVISED TASK ACQUISITION

An effective unsupervised meta-RL algorithm requires a method to acquire task distributions for an environment. We consider two concrete possibilities for such a procedure in this paper, though many other options are also possible for this stage.

**Task acquisition via random discriminators.**    A simple and surprisingly effective way to define arbitrary task distributions is to use random functions on states as reward functions. Given a uniformly distributed random variable $z \sim p(z)$, we define a random discriminator as a parametric function $D_{\phi_{rand}}(z|s)$, where the parameters $\phi_{rand}$ are chosen randomly (e.g., a random weight initialization for a neural network). The discriminator observes a state $s$ and outputs the probabilities for a categorical random variable $z$, hence drawing random decision boundaries in state space and giving us an appropriate random function. A reward function $r_z(s)$ can be extracted from this discriminator according as $\log(D_{\phi_{rand}}(z|s))$. This is not the only possible choice of random function, but we choose this so that we can more effectively compare to alternate task proposal mechanisms (described in the next paragraph). Note that this is not a random RL objective: the induced RL objective is affected by the inductive bias in the network and mediated by the CMP dynamics distribution. In our experiments, we find that random discriminators are able to acquire useful task distributions for simple tasks, but are not as effective as the tasks become more complicated.

**Task acquisition via diversity-driven exploration.**    We can acquire more varied tasks if we allow some amount of unsupervised environment interaction. We allow our agent to interact with the

environment, and propose a set of tasks unsupervised for meta-reinforcement learning. To do this, we consider a recently proposed method for unsupervised skill diversity method - Diversity is All You Need (DIAYN) (Eysenbach et al., 2018) for task acquisition. DIAYN attempts to acquire a set of behaviors that are distinguishable from one another, in the sense that they visit distinct states, while maximizing conditional policy entropy to encourage diversity (Haarnoja et al., 2018). Skills with high entropy that remain discriminable must explore a part of the state space far away from other skills. Formally, DIAYN learns a latent conditioned policy $\pi_\theta(a|s, z)$, with $z \sim p(z)$, where different values of $z$ induce different skills. The training process promotes discriminable skills by maximizing the mutual information between skills and states ($MI(s, z)$), while also maximizing the policy entropy $\mathcal{H}(a|s, z)$:

$$\mathcal{F}(\theta) \triangleq MI(s, z) + \mathcal{H}[a \mid s] - MI(a, z \mid s) = \mathcal{H}[a \mid s, z] + \mathcal{H}[z] - \mathcal{H}[z \mid s] \tag{1}$$

A learned discriminator $D_\phi(z|s)$ maximizes a variational lower bound on Equation 1 (see (Eysenbach et al., 2018) for proof). We train the discriminator to predict the latent variable $z$ from the observed state, and optimize the latent conditioned policy to maximize the log-likelihood of the discriminator correctly classifying states which are visited under different skills, while maximizing policy entropy. Under this formulation, we can think of the discriminator as *rewarding* the policy for producing discriminable skills, and the policy visitations as informing the training of the discriminator.

Given this procedure for learning useful skills without supervision, the natural question is how we can extract *tasks* from this that we can use for meta-learning, i.e sample suitable reward functions for each task. We can sample tasks using the discriminator learned by DIAYN by generating samples $z \sim p(z)$ and using the corresponding task reward $r_z(s) = \log(D_\phi(z|s))$. This reward is chosen since in DIAYN, the policy is reward according to $\log(D_\phi(z|s))$ to encourage discriminability of skills. Importantly, we are only using the discriminator as task reward for meta-learning, and not actually using the latent conditioned policy learned via DIAYN ($\pi_\theta(a|s, z)$). Compared to random discriminators, the tasks acquired by DIAYN are more likely to involve visiting diverse parts of the state space, potentially providing both a greater challenge to the corresponding policy and better coverage of the CMP's state space. This method is still fully unsupervised, as it requires no handcrafting of distance metrics or subgoals. It also does not require training a generative model of goals (Held et al., 2017).

### 3.3 META-REINFORCEMENT LEARNING WITH ACQUIRED TASK DISTRIBUTIONS

Once we have acquired a distribution of tasks, either randomly or through unsupervised exploration, we must choose a meta-learning algorithm to acquire the adaptation procedure from this task distribution. Which meta-learning algorithm is best suited for this problem? To formalize the *typical* meta-reinforcement learning problem, we assume that tasks $\tau \in \mathcal{T}$ are drawn from a manually specified task distribution $\tau_i \sim p(\tau)$, provided by the algorithm designer. These tasks are different MDPs. Each task $\tau_i$ is an MDP $M_i = (S, A, P, \gamma, R_i)$. The goal of meta-RL is to learn a reinforcement learning algorithm $f$ that can learn quickly on novel tasks drawn from $p(\tau)$. In contrast, in our problem setting we acquire the task distribution $p(\tau)$ completely *unsupervised*.

A particularly appealing choice for the meta-learning algorithm is model-agnostic meta-learning (Finn et al., 2017a), which trains a model that can adapt quickly to new tasks with standard gradient descent. In RL, this corresponds to the policy gradient, which means that $f$ simply runs policy gradient starting from a set of meta-learned initial parameters $\theta$. The meta-training objective for MAML is

$$\max_\theta \sum_{\tau_i \sim p(\tau)} \mathbb{E}_{\pi_{\theta'_i}} \left[ \sum_t R_i(s_t) \right] \quad \theta' = \theta + \alpha \mathbb{E}_{\pi_\theta} \left[ \sum_t R_i(s_t) \nabla_\theta \log \pi_\theta(a_t|s_t) \right]. \tag{2}$$

The rationale behind this objective is that, since the policy will be adapted at meta-test time to new tasks using policy gradient, we can optimize the policy parameters so that one step of policy gradient improves its performance on any meta-training task as much as possible. MAML learns a data-driven initialization that makes standard reinforcement learning fast on tasks drawn from the task distribution $p(\tau)$. Importantly, MAML uses standard RL via policy gradient to adapt to new tasks, ensuring that we can continuously keep improving on new tasks, even when those tasks lie outside the meta-training distribution. Prior work has observed that meta-learning with policy gradient improves extrapolation over meta-learners that learn the entire adaptation procedure (e.g.,

using a recurrent network (Finn & Levine, 2018)). Generalization to out-of-distribution samples is especially important for unsupervised meta-reinforcement learning methods because the actual task we might want to adapt to at meta-test time will almost certainly be out-of-distribution. For tasks that are too far outside of the meta-training set, MAML simply reverts to gradient-based RL. Other algorithms could also be used here, as discussed in the Section 3.5.

### 3.4 PRACTICAL ALGORITHM IMPLEMENTATION

A summary of a practical unsupervised meta-reinforcement learning algorithm is provided on the right. We first acquire a task distribution using unsupervised exploration (e.g., random discriminators or the DIAYN algorithm, as discussed in Section 3.2). We can sample from this task distribution by first sampling a random variable $z \sim p(z)$, and then use the reward induced by the resulting discriminator, $r_z(s) = \log(D_\phi(z|s))$ to update our policy. Having defined a procedure for sampling tasks, we perform gradient based meta-learning with MAML on this distribution until convergence. Convergence is decided by looking at the post-update reward for MAML and stopping when the difference between consecutive iterations is less than $\epsilon$. The resulting meta-learned policy is then able to adapt quickly to new tasks in the environment via standard policy gradient (Section 4) without requiring additional meta-training supervision.

### 3.5 WHICH UNSUPERVISED AND META-LEARNING PROCEDURES SHOULD WORK WELL?

Having introduced example instantiations of unsupervised meta-reinforcement learning, we discuss more generally what criteria each of the two procedures should satisfy - task acquisition and meta-reinforcement learning. What makes a good task acquisition procedure for unsupervised meta-reinforcement learning? Several criteria are desirable. First, we want the tasks that are learned to resemble the types of tasks that might be present at meta-test time. DIAYN receives no supervision in this regard, basing its task acquisition entirely on the dynamics of the CMP. A more guided approach could incorporate

---

**Algorithm 1:** Unsupervised Meta-Reinforcement Learning Pseudocode

**Data:** $\mathcal{M} \setminus R$, an MDP without a reward function
**Result:** a learning algorithm $f : D \to \pi$
Initialize $D = \emptyset$
$D_\phi \leftarrow \text{DIAYN}()$ or $D_\phi \leftarrow random$
**while** *not converged* **do**
 Sample latent task variables $z \sim p(z)$
 Extract corresponding task reward functions $r_z(s)$ using $D_\phi(z|s)$
 Update $f$ using MAML with reward $r_z(s)$

---

rate a limited number of human-specified tasks, or manually-provided guidance about valuable state space regions. Without any prior knowledge, we expect the ideal task distribution to induce a wide distribution over trajectories. As many distinct reward functions can have the same optimal policy, a random discriminator may actually result in a narrow distribution of optimal trajectories. In contrast, unsupervised task acquisition procedures like DIAYN, which mediate the task acquisition process via interactions with the environment (which imposes dynamically consistent structure on the tasks), are likely to yield better results than random task generation. The comparison to the random discriminator in our experiments sheds light on how a learned task distribution is important for this: while random and learned discriminators perform comparably on simple tasks, the learned discriminator performs better on more complex tasks.

In the absence of any mechanism that constraints the meta-training task distribution to resemble the meta-test distribution (which is unknown), we prefer methods that retain convergence guarantees, performing no worse than standard reinforcement learning algorithms that learn from scratch. Conveniently, gradient-based methods such as MAML gracefully revert to standard, convergent, reinforcement learning procedures on out-of-distribution tasks. Additionally, unlike methods which restrict the space for adaptation using latent conditioned policies such as DIAYN (Eysenbach et al., 2018), MAML does not lose policy expressivity because all policy parameters are being adapted, and often performs much better as seen in Fig 3.

We might then ask what kind of knowledge could possibly be "baked" into $f$ during meta-training. There are two sources of knowledge that can be acquired. First, a meta-learning procedure like MAML modifies the initial parameters $\theta$ of a policy $\pi_\theta(a|s)$. When $\pi_\theta(a|s)$ is represented by an expressive function class like a neural network, the initial setting of these parameters strongly affects how quickly the policy can be trained by gradient descent. Indeed, this is the rationale behind research

into more effective general-purpose initialization methods (Koturwar & Merchant, 2017; Xie et al.). Meta-training a policy essentially learns an effective weight initialization such that a few gradient steps can effectively modify the policy in functionally relevant ways.

The policy found by unsupervised meta-training also acquires an awareness of the dynamics of the given controlled Markov process (CMP). Intuitively, an ideal policy should adapt in the space of trajectories $\tau$, rather than the space of actions $a$ or parameters $\theta$; an RL update should modify the policy's *trajectory distribution*, which determines the reward function. Natural gradient algorithms impose equal-sized steps in the space of action distributions (Schulman et al., 2015), but this is not necessarily the ideal adaptation manifold, since systematic changes in output actions do not necessarily translate into systematic changes in trajectory or state distributions. In effect, meta-learning prepares the policy to modify its behavior in ways that cogently affect the states that are visited, which requires a parameter setting informed by the *dynamics* of the CMP. This can be provided effectively through unsupervised meta-reinforcement learning.

## 4 Experimental Evaluation

In our experiments, we aim to understand whether unsupervised meta-learning can accelerate reinforcement learning of new tasks. Whereas standard meta-learning requires a hand-specified task distribution at meta-training time, unsupervised meta-learning learns the task distribution through unsupervised interaction with the environment. A fair baseline that likewise uses requires no supervision is learning via RL from scratch without any meta-learning. As an upper bound, we include the unfair comparison to a standard meta-learning approach, where the meta-training distribution is manually designed. This method has access to a hand-specified task distribution that is not available to our method. We evaluate two variants of our approach: (a) task acquisition based on DIAYN followed by meta-learning using MAML, and (b) task acquisition using a randomly initialized discriminator followed by meta-learning using MAML. Our experiments aim to answer the following questions: (1) Does unsupervised meta-learning accelerate learning of unseen tasks? How does it compare to alternative approaches for exploration or skill discovery? (2) How does unsupervised meta-learning compare to meta-learning on a hand-specified task distribution? (3) When should unsupervised meta-learning with a learned task distribution be preferred over a meta-learning with a random discriminator? This last question sheds some light on the effect of task acquisition inductive bias on final reinforcement learning performance.

### 4.1 Tasks and Implementation Details

Our experiments study three simulated environments of varying difficulty: 2D point navigation, 2D locomotion using the "HalfCheetah," and 3D locomotion using the "Ant," with the latter two environments adapted from popular reinforcement learning benchmarks (Duan et al., 2016a). While the 2D navigation environment allows for direct control of position, HalfCheetah and Ant can only control their center of mass via feedback control with high dimensional actions (6D for HalfCheetah, 8D for Ant) and observations (17D for HalfCheetah, 111D for Ant).

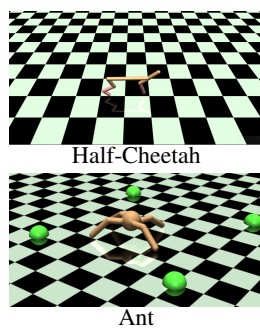

Half-Cheetah

Ant

The evaluation tasks, shown in Figure 6, are similar to prior work (Finn et al., 2017a; Pong et al., 2018): 2D navigation and ant require navigating to goal positions, while the half cheetah must run at different goal velocities. These tasks are not accessible to our algorithm during meta-training.

We used the default hyperparameters for MAML across all tasks, varying the meta-batch size according to the number of skills that the discriminator is parameterized by - 50 for pointmass, and 20 for cheetah and ant. We found that the default architecture - 2 layer MLP with 300 units each and ReLU non-linearities worked quite well for meta-training. We also used the default hyperparameters for DIAYN to acquire skills. We swept over learning rates for learning from scratch via vanilla policy gradient, and found that using ADAM with adaptive step size is the most stable and quick at learning.

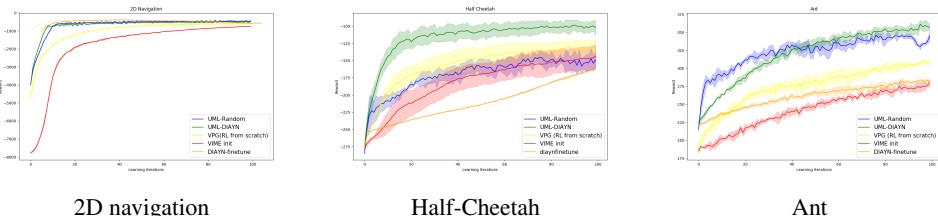

| 2D navigation | Half-Cheetah | Ant |
|:---:|:---:|:---:|

Figure 3: **Unsupervised meta-learning accelerates learning**: After unsupervised meta-learning, our approach (UML-DIAYN and UML-RANDOM) quickly learns a new task significantly faster than learning from scratch, especially on complex tasks. Learning the task distribution with DIAYN helps more for complex tasks. Results are averaged across 20 evaluation tasks, and 3 random seeds for testing. UML-DIAYN and random also significantly outperform learning with DIAYN initialization or an initialization with a policy pretrained with VIME.

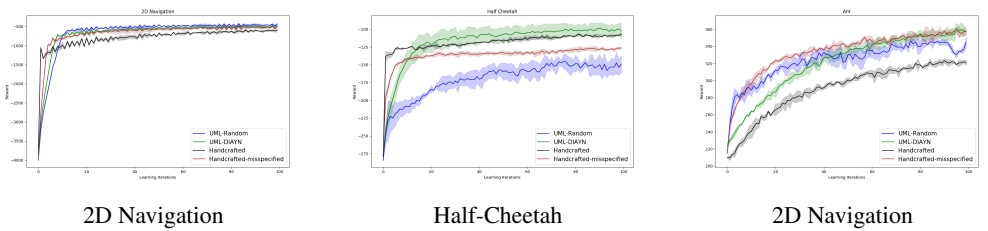

| 2D Navigation | Half-Cheetah | 2D Navigation |
|:---:|:---:|:---:|

Figure 4: **Comparison with handcrafted tasks**: Unsupervised meta-learning (UML-DIAYN) is competitive with meta-training on handcrafted reward functions (i.e., an oracle). A misspecified, handcrafted meta-training task distribution often performs worse, illustrating the benefits of learning the task distribution.

## 4.2 FAST ADAPTATION AFTER UNSUPERVISED META LEARNING

The comparison between the two variants of unsupervised meta-learning and learning from scratch is shown in Fig 3, and we compare to hand-crafted task distributions in Fig 4. We also compare with 2 additional baselines in Figure 3 - pretraining with VIME Houthooft et al. (2016) and DIAYN + finetuning. In the pre-training with VIME, we run VIME, which is a novelty based exploration method, for a number of iterations with no reward such that it simply learns to explore. This initializaiton is then used for finetuning on actual task reward. In the DIAYN + finetuning comparison we follow the procedure in Eysenbach et al. (2018) and first train DIAYN, following which we finetune the learned policy by first choosing the highest reward skill for the current task, fixing the skill and then finetuning the policy with task reward.

We observe in all cases that the UML-DIAN variant of unsupervised meta-learning produces an RL procedure that outperforms reinforcement learning from scratch, VIME-init and DIAYN + finetuning, suggesting that unsupervised interaction with the environment and meta-learning is effective in producing environment-specific but task-agnostic priors that accelerate learning on new, previously unseen tasks. The comparison with VIME shows that the speed of learning is not just about exploration but is indeed about fast adaptation. The comparison with DIAYN + finetuning indicates the benefit of actually training the policy for fast adaptation on these tasks, rather than directly relying on the initial skills to be good, without optimizing for fast adaptation explicitly. While DIAYN + finetuning learns very quickly on pointmass, it is unable to perform as well on cheetah and ant despite starting off at a higher initial reward. This baseline likely performs badly because the actual learned skills weren't that similar to the testing task distribution. In our experiments thus far UML-DIAYN always performs better than learning from scratch, although the benefit varies across tasks depending on the actual performance of DIAYN. We expect that unless DIAYN proposes a very biased set of tasks, UML-DIAYN will likely do better than learning from scratch in most cases.

Interestingly, in many cases (in Fig 4) the performance of unsupervised meta-learning with DIAYN matches or exceeds that of the hand-designed task distribution. We see that on the 2D navigation task, while handcrafted meta-learning is able to learn very quickly initially, it performs similarly after 100 steps. For the cheetah environment as well, handcrafted meta-learning is able to learn very quickly to start off, but is quickly matched by unsupervised meta-RL with DIAYN. We also see on

the HalfCheetah that, if we meta-test using an initialization learned with a slightly different task distribution, performance degrades to below that of our approach. This result indicates that unsupervised environment interaction can extract a sufficiently diverse set of tasks to make unsupervised meta-learning useful. However on the ant task, we see that hand-crafted meta-learning does do better than UML-DIAYN, likely because the task distribution is more challenging, and a better unsupervised task proposal algorithm would improve the performance of a meta-learner.

The comparison between the two unsupervised meta-learning variants is also illuminating: while the DIAYN-based variant of our method generally achieves the best performance, even the random discriminator is able to provide a sufficient diversity of tasks to produce meaningful acceleration over learning from scratch in the case of 2D navigation and ant.

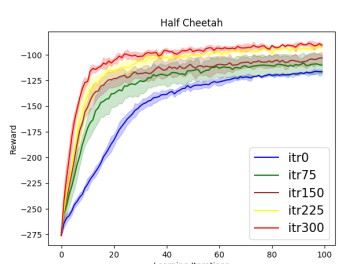

Figure 5: Analysis of effect of additional meta-training on meta-test time learning of new tasks. For larger iterations of meta-trained policies, we have improved test time performance, showing that additional meta-training is beneficial.

This result has two interesting implications. First, it suggests that unsupervised meta-learning is an effective tool for learning an environment prior, even when the meta-training task distribution does not necessarily broadly cover the state space. Although the performance of unsupervised meta-learning can be improved with better coverage using DIAYN (as seen in Fig 3), even the random discriminator version provides competitive advantages over learning from scratch. Second, the comparison provides a clue for identifying the source of the structure learned through unsupervised meta-learning: though the particular task distribution has an effect on performance, simply interacting with the environment (without structured objectives, using a random discriminator) already allows meta-RL to learn effective adaptation strategies in a given environment. That is, the performance cannot be explained only by the unsupervised procedure (DIAYN) capturing the right task distribution.

To understand the method performance more clearly, we also add an ablation study where we compare the meta-test performance of policies at different iterations along meta-training. This shows the effect that additional meta-training has on the fast learning performance for new tasks. This comparison is shown in Fig 5. As can be seen here, at iteration 0 of meta-training the policy is not a very good initialization for learning new tasks. As we move further along the meta-training process, we see that the meta-learned initialization becomes more and more effective at learning new tasks. This shows a clear correlation between additional meta-training and improved meta test-time performance.

## 4.3 ANALYSIS OF LEARNED TASK DISTRIBUTIONS

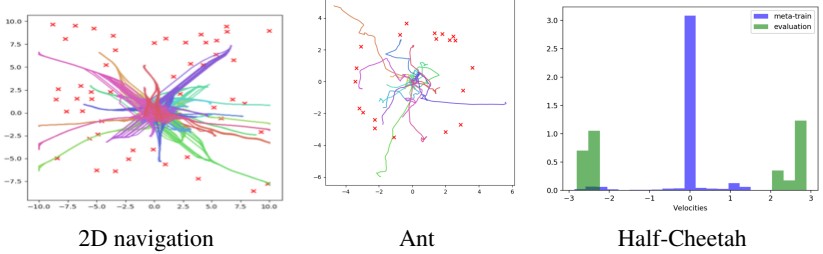

| 2D navigation | Ant | Half-Cheetah |

Figure 6: **Learned meta-training task distribution and evaluation tasks**: We plot the center of mass for various skills discovered by point mass and ant using DIAYN, and a blue histogram of goal velocities for cheetah. Evaluation tasks, which are not provided to the algorithm during meta-training, are plotted as red 'x' for ant and pointmass, and as a green histogram for cheetah. While the meta-training distribution is broad, it does not fully cover the evaluation tasks. Nonetheless, meta-learning on this *learned* task distribution enables efficient learning on a test task distribution.

We can analyze the tasks discovered through unsupervised exploration and compare them to tasks we evaluate on at meta-test time. Figure 6 illustrates these distributions using scatter plots for 2D

navigation and the Ant, and a histogram for the HalfCheetah. Note that we visualize dimensions of the state that are relevant for the evaluation tasks – positions and velocities – but these dimensions are *not* specified in any way during unsupervised task acquisition, which operates on the entire state space. Although the tasks proposed via unsupervised exploration provide fairly broad coverage, they are clearly quite distinct from the meta-test tasks, suggesting the approach can tolerate considerable distributional shift. Qualitatively, many of the tasks proposed via unsupervised exploration such as jumping and falling that are not relevant for the evaluation tasks. Our choice of the evaluation tasks was largely based on prior work, and therefore not tailored to this exploration procedure. The results for unsupervised meta-reinforcement learning therefore suggest quite strongly that unsupervised task acquisition can provide an effective meta-training set, at least for MAML, even when evaluating on tasks that do not closely match the discovered task distribution.

## 5 Discussion and Future Work

We presented an unsupervised approach to meta-reinforcement learning, where meta-learning is used to acquire an efficient reinforcement learning procedure without requiring hand-specified task distributions for meta-training. This approach accelerates RL without relying on the manual supervision required for conventional meta-learning algorithms. Our experiments indicate that unsupervised meta-RL can accelerate learning on a range of tasks, outperforming learning from scratch and often matching the performance of meta-learning from hand-specified task distributions.

There are important caveats to this discussion of unsupervised meta-learning. While proposing tasks via DIAYN or random discriminators may be effective in these domains, there are domains where such task proposal mechanisms may not be as effective. The performance of UML critically depends on this and is likely to suffer. Additionally, it is important to note the additional sample complexity that comes from using DIAYN first to acquire skills followed by an on-policy meta-RL algorithm such as MAML. Moving to more efficient RL algorithms and training UML in the loop would be a good step towards improving the efficiency of such algorithms.

In this work, we studied the case where unsupervised meta-learning is performed under the same dynamics, and the aim is to learn a learning procedure that succeeds for other objectives in the same environment. A natural extension of our approach would be to perform unsupervised meta-learning on a variety of different environments, which would provide for even wider generalization, and this would be an exciting direction for future work.

As our work is the first foray into unsupervised meta-learning, our approach opens a number of questions about unsupervised meta-learning algorithms. While we focus on purely unsupervised task proposal mechanisms, it is straightforward to incorporate minimally-informative priors into this procedure. For example, we might restrict the learned reward functions to operate on only part of the state. We consider the reinforcement learning setting in our work because environment interaction mediates the unsupervised learning process, ensuring that there is something to learn even without access to task reward. An interesting direction to study in future work is the extension of unsupervised meta-learning to domains such as supervised classification, which might hold the promise of developing new unsupervised learning procedures powered by meta-learning.

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
