# OpenReview forum: "Unsupervised Meta-Learning for Reinforcement Learning"
_ICLR.cc/2019/Conference_

### Official Review · AnonReviewer3 · 2018-10-31
**An interresting but rushed paper**

**Rating:** 4
**Confidence:** 2

**Review:**

*Summary:* The present paper proposes to use  Model Agnostic Meta Learning to (meta)-learn in an unsupervised fashion the reward function of a Markov decision process in the context of Reinforcement Learning (RL). The distribution of tasks corresponds to a distribution of reward functions which are created thanks to random discriminators or diversity driven exploration.

*Clarity:* The goal is well stated but the presentation of the method is confusing.

There is a  constant switch between caligraphic and roman D. Could you homogenize the notations?

Could you keep the same notation for the MDP (eg in the introduction and 3.5, the discount factor disappeared)

In the introduction the learning algorithm takes a MDP in mathcal{T} and return a policy. In the remaining of the paper mathcal{D} is used. Could you clarify? I guess this is because only the reward of the MDP is meta-learned, which is itself based on D_phi?

you choose r = log(Discriminator). Could you explain this choice? Is there alternative choices?

In subsection 3.4, why the p in the reward equation?

Algorithm 1 is not clear at all and needs to be rewritten:
   - Could you specify the stopping criterion for MAML you used?
   - Could you number the steps of the algorithm?

Concerning the experiments:

In my opinion the picture of the dataset ant and cheeta is irrelevant and could be removed for more explainations of the method.

It would be very nice to have color-blind of black and white friendly graphs.

In the abstract, I don't think the word demonstrate should be used about the experimental result. As pointed out in the experimental section the experiment are here rather to give additional insight on why and when the proposed method works well.

Your method learns faster than RL from scratch on the proposed dataset in terms of iteration. What about monitoring the reward in terms of time, including the meta-learning step. Is there any important constant overhead in you the proposed method? How does the meta training time impact the training time? Do you have examples of datasets where the inductive bias is not useful or worst than RL from scratch? If yes could you explain why the method is not as good as RL from scratch?

The presentation of the result is weird.
Why figure 4 does not include the ant dataset? Why no handcrafted misspecified on 2D navigation?
Figure 3 and 4 could be merged since many curves are in common.

How have you tuned your hyperparameters of each methods? Could you put in appendix the exact protocol you used, specifying the how hyperparameters of the whole procedured are chosen, what stopping criterion are used, for the sake of reproducibility. A an internet link to a code repository used to produce the graphs would be very welcome in the final version if accepted.

In the conclusion, could you provide some of the questions raised?

*Originality and Significance:* As I'm not an expert in the field, it is difficult for me to evaluate the interest of the RL comunity in this work. Yet to the best of my knowledge the work presented is original, but the lack of clarity and ability to reproduce the results might hinder the impact of the paper.

Typos:
Eq (2) missing a full stop
Missing capital at ´´ update´´  in algorithm 1
End of page 5, why the triple dots?

---

### Official Review · AnonReviewer1 · 2018-11-05

**Rating:** 6
**Confidence:** 3

**Review:**

The authors propose a framework for unsupervised meta-reinforcement learning. This aims to perform meta-learning in reinforcement learning context without an specification of the meta tasks being pre-specified. The authors propose two algorithms to acquire the task distributions (unsupervised). In particular the better performing approach relies on the recently introduced DIAYN algorithm.  Experiments are presented on several simple benchmark datasets.

The authors propose an interesting formulation of a useful problem: finding tasks automatically that aid meta-learning. To the best of my knowledge this is indeed a novel idea and indeed an important one. On the other hand the authors only take relatively early steps towards solving this task and the discussion of what is a good unsupervised task selection is underwhelming. Indeed one is not left of a clear idea of what kind of inductive biases would be a valid approach to this problem and why the authors consider specifically the two approaches described.

For the experiments it seems a lot of the key improvements come from the DIAYN algorithm. The experiments are also presented on relatively toy tasks and mainly compare to RL from scratch approaches.  It would be interesting to see the effectiveness of these methods on harder problems. For the experiments I would be interested to know if one could compare directly to using DIAYN as in the original Eysenbach et al for example as an initialization.

Overall the paper presents several interesting results and I think the high level formulation could have some potential impacts, although the limits  of such an approach are not completely clear and whether it can be effective on complex tasks is not fully known yet.

---

### Official Review · AnonReviewer5 · 2018-11-13
**Both the exposition and the formulation leave much to be desired**

**Rating:** 3
**Confidence:** 4

**Review:**

The paper considers a particular setting of so-called meta-reinforcement learning (meta-RL) where there is a distribution over reward functions (the transition function is fixed) and with some access to this distribution, the goal is to produce a learning algorithm that  "learns well" on the distribution. At training time, a sample of reward functions are drawn and an algorithm returns a learning program that at test time, can reinforcement learn the test environment as specified by a test reward function. The paper proposes to generate a set of training reward functions instead of relying on some "manual" specification, thus the term "unsupervised" in the title. It also proposes an algorithm, basing on a recent work in skill discovery (DIAYN, not yet peer-reviewed), to find such reward functions.

Firstly, the exposition is hard to follow. For example, the "Task acquisition via random discriminators" subsection, without first mentioning DIAYN, seems out-of-context: what is D_phi_rand(z|s)? a joint distribution of (reward function, state) makes no sense. It only makes sense when there is a stochastic process, e.g. MDP coupled with a policy.

Secondly, the reasoning is very informal based on a vague vocabulary (not trying to pick on the authors, these are not uncommon in deep learning literature) are used without rigorous arguments. Section 3.1 brought up a natural objection -- I applaud the authors' self-critique -- based on "no free lunch theorem," but it dismisses it via "the specific choice for the unsupervised learning procedure and meta-learning algorithm can easily impose an inductive bias" without specifying what (and how) choice leads to what "inductive biases." This is crucial as the authors seem to suggest that although the "inductive bias" is important -- task design expresses such -- an unsupervised method, which requires no supervision, can do as well.

Thirdly, the baseline comparison seems inappropriate to me. The "fair" baseline the authors proposed was to RL a test task from scratch. But this is false as the meta-RL agent enjoys access to the transition dynamics (controlled Markov process, CMP in the paper) during the so-called meta-training (before meta-testing on the test task). In fact, a more appropriate baseline would be initialize an RL agent with with a correct model (if sample complexity in training is not a concern, which seems to be the case as it was never addressed in the paper) or a model estimated from random sample transitions (if we are mindful of sample complexity which seems more reasonable to me). One may object that a (vanilla) policy-gradient method cannot incorporate an environment model but why should we restrict ourselves to these model-free methods in this setting where the dynamics can be accessed during (meta-)training?

Pros:
1. It connects skill discovery and meta-RL. Even though such connection was not made explicitly clear in the writing, its heavy reliance on a recent (not yet peer-reviewed) paper suggests such. It seems to want to suggest it through a kind of "duality" between skills/policies and rewards/tasks (z in the paper denotes the parameter of a reward function and also the parameter of policy). But are there any difference between the two settings?

Cons:
1. The writing is imprecise and often hard to follow.
2. The setting considered is not well motivated. How does an unsupervised method provide the task distribution before seeing any tasks?
3. The restriction of tasks to different reward functions made the proposed baseline seem unfairly weak.

In summary, I could not recommend accepting this work as it stands. I sincerely hope that the authors will be more precise in their future writing and focus on articulating and testing their key hypotheses.

---

### Meta-Review · Area_Chair1 · 2018-12-13
**Improvement needed**

**Confidence:** 5
**Recommendation:** Reject

**Metareview:**

This paper introduces unsupervised meta-learning algorithms for RL. Major concerns of the paper include: 1. Lack of clarity. The presentation of the method can be improved. 2. The motivation and justification of applying unsupervised meta-learning needs to be strengthened. More discussions and better motivating examples may be useful. 3. Experimental details are not sufficient and comparisons may not be sufficient to support the aim. Overall, this paper cannot be accepted yet.